# Repulsive Deep Ensembles are Bayesian

**Francesco D'Angelo**
ETH Zürich
Zürich, Switzerland
dngfra@gmail.com

**Vincent Fortuin**
ETH Zürich
Zürich, Switzerland
fortuin@inf.ethz.ch

## Abstract

Deep ensembles have recently gained popularity in the deep learning community for their conceptual simplicity and efficiency. However, maintaining functional diversity between ensemble members that are independently trained with gradient descent is challenging. This can lead to pathologies when adding more ensemble members, such as a saturation of the ensemble performance, which converges to the performance of a single model. Moreover, this does not only affect the quality of its predictions, but even more so the uncertainty estimates of the ensemble, and thus its performance on out-of-distribution data. We hypothesize that this limitation can be overcome by discouraging different ensemble members from collapsing to the same function. To this end, we introduce a kernelized repulsive term in the update rule of the deep ensembles. We show that this simple modification not only enforces and maintains diversity among the members but, even more importantly, transforms the maximum a posteriori inference into proper Bayesian inference. Namely, we show that the training dynamics of our proposed repulsive ensembles follow a Wasserstein gradient flow of the KL divergence to the true posterior. We study repulsive terms in weight and function space and empirically compare their performance to standard ensembles and Bayesian baselines on synthetic and real-world prediction tasks.

## 1  Introduction

There have been many recent advances on the theoretical properties of sampling algorithms for approximate Bayesian inference, which changed our interpretation and understanding of them. Particularly worth mentioning is the work of Jordan et al. [38], who reinterpret Markov Chain Monte Carlo (MCMC) as a gradient flow of the KL divergence over the Wasserstein space of probability measures. This new formulation allowed for a deeper understanding of approximate inference methods but also inspired the inception of new and more efficient inference strategies. Following this direction, Liu and Wang [51] recently proposed the Stein Variational Gradient Descent (SVGD) method to perform approximate Wasserstein gradient descent. Conceptually, this method, which belongs to the family of particle-optimization variational inference (POVI), introduces a repulsive force through a kernel acting in the parameter space to evolve a set of samples towards high-density regions of the target distribution without collapsing to a point estimate.

Another method which has achieved great success recently are ensembles of neural networks (so-called *deep ensembles*), which work well both in terms of predictive performance [42, 80] as well as uncertainty estimation [65], and have also been proposed as a way to perform approximate inference in Bayesian neural networks [82, 36]. That being said, while they might allow for the averaging of predictions over several hypotheses, they do not offer any guarantees for the diversity between those hypotheses nor do they provably converge to the true Bayesian posterior under any meaningful limit. In this work, we show how the introduction of a repulsive term between the members in the ensemble, inspired by SVGD, not only naïvely guarantees the diversity among the members, avoiding their

35th Conference on Neural Information Processing Systems (NeurIPS 2021).

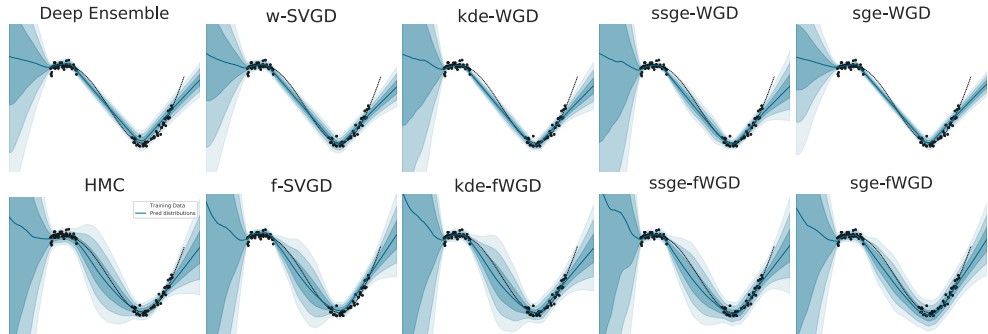

Figure 1: **BNN 1D regression.** The function-space methods (SVGD and WGD) approach the HMC posterior more closely, while the standard deep ensembles and weight-space methods fail to properly account for the uncertainty, especially the in-between uncertainty.

collapse in parameter space, but also allows for a reformulation of the method as a gradient flow of the KL divergence in the Wasserstein space of distributions. It thus allows to endow deep ensembles with convergence guarantees to the true Bayesian posterior.

An additional problem is that BNN inference in weight space can lead to degenerate solutions, due to the overparametrization of these models. That is, several samples could have very different weights but map to the same function, thus giving a false sense of diversity in the ensemble. This property, that we will refer to as *non-identifiability* of neural networks (see Appendix A), can lead to redundancies in the posterior distribution. It implies that methods like MCMC sampling, deep ensembles, and SVGD waste computation in local modes that account for equivalent functions. Predictive distributions approximated using samples from these modes do not improve over a simple point estimate and lead to a poor uncertainty estimation. Following this idea, Wang et al. [76] introduced a new method to extend POVI methods to function space, overcoming this limitation. Here, we also study an update rule that allows for an approximation of the gradient flow of the KL divergence in function space in our proposed repulsive ensembles.

We make the following contributions:

- We derive several different repulsion terms that can be added as regularizers to the gradient updates of deep ensembles to endow them with Bayesian convergence properties.

- We show that these terms approximate Wasserstein gradient flows of the KL divergence and can be used both in weight space and function space.

- We compare these proposed methods theoretically to standard deep ensembles and SVGD and highlight their different guarantees.

- We assess all these methods on synthetic and real-world deep learning tasks and show that our proposed repulsive ensembles can achieve competitive performance and improved uncertainty estimation.

## 2   Repulsive Deep Ensembles

In supervised deep learning, we typically consider a likelihood function $p(\boldsymbol{y}|f(\boldsymbol{x};\mathbf{w}))$ (e.g., Gaussian for regression or Categorical for classification) parameterized by a neural network $f(\boldsymbol{x};\mathbf{w})$ and training data $\mathcal{D} = \{(\boldsymbol{x}_i, \boldsymbol{y}_i)\}_{i=1}^n$ with $\boldsymbol{x} \in \mathcal{X}$ and $\boldsymbol{y} \in \mathcal{Y}$. In Bayesian neural networks (BNNs), we are interested in the posterior distribution of all likely networks given by $p(\mathbf{w}|\mathcal{D}) \propto \prod_{i=1}^n p(\boldsymbol{y}_i|f(\boldsymbol{x}_i;\mathbf{w}))\, p(\mathbf{w})$, where $p(\mathbf{w})$ is the prior distribution over weights. Crucially, when making a prediction on a test point $\boldsymbol{x}^*$, in the Bayesian approach we do not only use a single parameter $\widehat{\mathbf{w}}$ to predict $\boldsymbol{y}^* = f(\boldsymbol{x}^*; \widehat{\mathbf{w}})$, but we marginalize over the whole posterior, thus taking all possible explanations of the data into account:

$$p(\boldsymbol{y}^*|\boldsymbol{x}^*, \mathcal{D}) = \int p(\boldsymbol{y}^*|f(\boldsymbol{x}^*;\mathbf{w}))\, p(\mathbf{w}|\mathcal{D})\, \mathrm{d}\mathbf{w} \tag{1}$$

While approximating the posterior of Bayesian neural networks (or sampling from it) is a challenging task, performing maximum a posteriori (MAP) estimation, which corresponds to finding the mode of the posterior, is usually simple. Ensembles of neural networks use the non-convexity of the MAP optimization problem to create a collection of $K$ independent—and possibly different— solutions. Considering $n$ weight configurations of a neural network $\{\mathbf{w}_i\}_{i=1}^n$ with $\mathbf{w}_i \in \mathbb{R}^d$, the dynamics of the ensemble under the gradient of the posterior lead to the following update rule at iteration $t$:

$$\mathbf{w}_i^{t+1} \leftarrow \mathbf{w}_i^t + \epsilon_t \phi(\mathbf{w}_i^t)$$
$$\text{with} \qquad \phi(\mathbf{w}_i^t) = \nabla_{\mathbf{w}_i^t} \log p(\mathbf{w}_i^t | \mathcal{D}) \,, \tag{2}$$

with step size $\epsilon_t$. Ensemble methods have a long history [e.g., 45, 26, 6] and were recently revisited for neural networks [42] and coined *deep ensembles*. The predictions of the different members are combined to create a predictive distribution by using the solutions to compute the Bayesian model average (BMA) in Eq. (1). Recent works [65] have shown that deep ensembles can outperform some of the Bayesian approaches for uncertainty estimation. Even more recently, Wilson and Izmailov [82] argued that deep ensembles can be considered a compelling approach to Bayesian model averaging. Despite these ideas, the ability of deep ensembles to efficiently average over multiple hypotheses and to explore the functional landscape of the posterior distribution studied in [18] does not guarantee sampling from the right distribution. Indeed, the additional Langevin noise introduced in [77], which is not considered in deep ensembles, is crucial to ensure samples from the true Bayesian posterior.

From a practical standpoint, since the quality of an ensemble hinges on the diversity of its members, many methods were recently proposed to improve this diversity without compromising the individual accuracy. For instance, Wenzel et al. [80] propose hyper-deep ensembles that combine deep networks with different hyperparameters. Similarly, cyclical learning-rate schedules can explore several local minima for the ensemble members [33]. Alternatively, Rame and Cord [67] proposed an information-theoretic framework to avoid redundancy in the members and Oswald et al. [63] studied possible interactions between members based on weight sharing. However, the absence of a constraint that prevents particles from converging to the same mode limits the possibility of improvement by introducing more ensemble members. This means that any hopes to converge to different modes must exclusively rely on:

1. the randomness of the initialization
2. the noise in the estimation of the gradients due to minibatching
3. the number of local optima that might be reached during gradient descent.

Moreover, the recent study of Geiger et al. [25] showed how the empirical test error of the ensemble converges to the one of a single trained model when the number of parameters goes to infinity, leading to deterioration of the performance. In other words, the bigger the model, the harder it is to maintain diversity in the ensemble and avoid collapse to the same solution. This is intuitively due to the fact that bigger models are less sensitive to the initialization. Namely, in order for them to get stuck in a local minimum, they must have second derivatives that are positive simultaneously in all directions. As the number of hidden units gets larger, this becomes less likely.

## 2.1 Repulsive force in weight space

To overcome the aforementioned limitations of standard deep ensembles, we introduce, inspired by SVGD [51], a deep ensemble with members that interact with each other through a repulsive component. Using a kernel function to model this interaction, the single models repel each other based on their position in the weight space, so that two members can never assume the same weights. Considering a stationary kernel $k(\cdot, \cdot) : \mathbb{R}^d \times \mathbb{R}^d \to \mathbb{R}$ acting in the parameter space of the neural network, a repulsive term $\mathcal{R}$ can be parameterized through its gradient:

$$\phi(\mathbf{w}_i^t) = \nabla_{\mathbf{w}_i^t} \log p(\mathbf{w}_i^t | \mathcal{D}) - \mathcal{R}\left(\left\{\nabla_{\mathbf{w}_i^t} k(\mathbf{w}_i^t, \mathbf{w}_j^t)\right\}_{j=1}^n\right) \,. \tag{3}$$

To get an intuition for the behavior of this repulsive term and its gradients, we can consider the RBF kernel $k(\mathbf{w}_i, \mathbf{w}_j) = \exp\left(-\frac{1}{h}||\mathbf{w}_i - \mathbf{w}_j||^2\right)$ with lengthscale $h$ and notice how its gradient

$$\nabla_{\mathbf{w}_i^t} k(\mathbf{w}_i^t, \mathbf{w}_j^t) = \frac{2}{h}(\mathbf{w}_j^t - \mathbf{w}_i^t) k(\mathbf{w}_i^t, \mathbf{w}_j^t)$$

drives $\mathbf{w}_i$ away from its neighboring members $\mathbf{w}_j$, thus creating a repulsive effect. Naturally, not all the choices of $\mathcal{R}$ induce this effect. One of the simplest formulations to obtain it is via a linear combination of the kernel gradients scaled by a positive factor, that is, $\beta \sum_{j=1}^{n} \nabla_{\mathbf{w}_i^t} k(\mathbf{w}_i^t, \mathbf{w}_j^t)$ with $\beta \in \mathbb{R}_*^+$. We will see in Section 3 how the choice of $\beta$ can be justified in order to obtain convergence to the Bayesian posterior together with alternative possible formulations of $\mathcal{R}$ that preserve this convergence.

## 2.2 Repulsive force in function space

To overcome the aforementioned overparameterization issue, the update in Eq. (3) can be formulated in function space instead of weight space. Let $\boldsymbol{f} : \mathbf{w} \mapsto f(\cdot\,; \mathbf{w})$ be the map that maps a configuration of weights $\mathbf{w} \in \mathbb{R}^d$ to the corresponding neural network regression function and denote as $\boldsymbol{f}_i := f(\cdot\,; \mathbf{w}_i)$ the function with a certain configuration of weights $\mathbf{w}_i$. We can now consider $n$ particles in function space $\{\boldsymbol{f}_i\}_{i=1}^n$ with $\boldsymbol{f} \in \mathcal{F}$ and model their interaction with a general positive definite kernel $k(\cdot, \cdot)$. We also consider the implicit functional likelihood $p(\boldsymbol{y}|\boldsymbol{x}, \boldsymbol{f})$, determined by the measure $p(\boldsymbol{y}|\boldsymbol{x}, \mathbf{w})$ in the weight space, as well as the functional prior $p(\boldsymbol{f})$, which can either be defined separately (e.g., using a GP) or modeled as a push-forward measure of the weight-space prior $p(\mathbf{w})$. Together, they determine the posterior in function space $p(\boldsymbol{f}|\mathcal{D})$. The functional evolution of a particle can then be written as:

$$\boldsymbol{f}_i^{t+1} \leftarrow \boldsymbol{f}_i^t + \epsilon_t \phi(\boldsymbol{f}_i^t)$$

$$\text{with} \qquad \phi(\boldsymbol{f}_i^t) = \nabla_{\boldsymbol{f}_i^t} \log p(\boldsymbol{f}_i^t|\mathcal{D}) - \mathcal{R}\left(\left\{\nabla_{\boldsymbol{f}_i^t} k(\boldsymbol{f}_i^t, \boldsymbol{f}_j^t)\right\}_{j=1}^n\right). \qquad (4)$$

However, computing the update in function space is neither tractable nor practical, which is why two additional considerations are needed. The first one regards the infinite dimensionality of function space, which we circumvent using a canonical projection into a subspace:

**Definition 1** (Canonical projection). *For any $A \subset \mathcal{X}$, we define $\pi_A : \mathbb{R}^{\mathcal{X}} \to \mathbb{R}^A$ as the canonical projection onto $A$, that is, $\pi_A(f) = \{f(a)\}_{a \in A}$.*

In other words, the kernel will not be evaluated directly in function space, but on the projection $k\big(\pi_B(f), \pi_B(f')\big)$, with $B$ being a subset of the input space given by a batch of training data points. The second consideration is to project this update back into the parameter space and evolve a set of particles there, because ultimately we are interested in representing the functions by parameterized neural networks. For this purpose, we can use the Jacobian of the $i$-th particle as a projector:

$$\phi(\mathbf{w}_i^t) = \left(\frac{\partial \boldsymbol{f}_i^t}{\partial \mathbf{w}_i^t}\right)^\top \left[\nabla_{\boldsymbol{f}_i^t} \log p(\boldsymbol{f}_i^t|\mathcal{D}) - \mathcal{R}\left(\left\{\nabla_{\boldsymbol{f}_i^t} k(\pi_B(\boldsymbol{f}_i^t), \pi_B(\boldsymbol{f}_j^t))\right\}_{j=1}^n\right)\right]. \qquad (5)$$

## 2.3 Comparison to Stein variational gradient descent

Note that our update is reminiscent of SVGD [51], which in parameter space can be written as:

$$\phi(\mathbf{w}_i^t) = \sum_{j=1}^n k(\mathbf{w}_i^t, \mathbf{w}_j^t) \nabla_{\mathbf{w}_i^t} \log p(\mathbf{w}_i^t|\mathcal{D}) + \sum_{j=1}^n \nabla_{\mathbf{w}_j^t} k(\mathbf{w}_j^t, \mathbf{w}_i^t). \qquad (6)$$

It is important to notice that here, the gradients are averaged across all the particles using the kernel matrix. Interestingly, SVGD can be asymptotically interpreted as gradient flow of the KL divergence under a new metric induced by the Stein operator [16, 50] (see Appendix D for more details). Moving the inference from parameter to function space [76] leads to the update rule

$$\phi(\mathbf{w}_i^t) = \left(\frac{\partial \boldsymbol{f}_i^t}{\partial \mathbf{w}_i^t}\right)^\top \left(\frac{1}{n}\sum_{j=1}^n k(\boldsymbol{f}_i^t, \boldsymbol{f}_j^t) \nabla_{\boldsymbol{f}_j^t} \log p(\boldsymbol{f}_j^t|\mathcal{D}) + \nabla_{\boldsymbol{f}_j^t} k(\boldsymbol{f}_i^t, \boldsymbol{f}_j^t)\right). \qquad (7)$$

This way of averaging gradients using a kernel can be dangerous in high-dimensional settings, where kernel methods often suffer from the curse of dimensionality. Moreover, in Eq. (6), the posterior gradients of the particles are averaged using their similarity in weight space, which can be misleading in multi-modal posteriors. Worse yet, in Eq. (7), the gradients are averaged in function space and

are then projected back using exclusively the $i$-th Jacobian, which can be harmful given that it is not guaranteed that distances between functions evaluated on a subset of their input space resemble their true distance. Our proposed method, on the other hand, does not employ any averaging of the posterior gradients and thus comes closest to the true particle gradients in deep ensembles.

# 3 Repulsive deep ensembles are Bayesian

So far, we represented the repulsive force as a general function of the gradients of a kernel. In this section, we show how to determine the explicit form of the repulsive term, such that the resulting update rule is equivalent to the discretization of the gradient flow dynamics of the KL divergence in Wasserstein space. We begin by introducing the concepts of particle approximation and gradient flow.

## 3.1 Particle approximation

A particle-based approximation of a target measure depends on a set of weighted samples $\{(x_i, w_i)\}_{i=1}^n$, for which an empirical measure can be defined as

$$\rho(x) = \sum_{i=1}^n w_i \, \delta(x - x_i) \,, \tag{8}$$

where $\delta(\cdot)$ is the Dirac delta function and the weights $w_i$ satisfy $w_i \in [0, 1]$ and $\sum_{i=1}^n w_i = 1$. To approximate a target distribution $\pi(x)$ using the empirical measure, the particles and their weights need to be selected in a principled manner that minimizes some measure of distance between $\pi(x)$ and $\rho(x)$ (e.g., a set of $N$ samples with weights $w_i = 1/N$ obtained using an MCMC method).

## 3.2 Gradient flow in parameter space

Given a smooth function $J : \mathbb{R}^d \to \mathbb{R}$ in Euclidean space, we can minimize it by creating a path that follows its negative gradient starting from some initial conditions $x_0$. The curve $x(t)$ with starting point $x_0$ described by that path is called *gradient flow*. The dynamics and evolution in time of a considered point in the space under this minimization problem can be described as the ODE[1]

$$\frac{dx}{dt} = -\nabla J(x) \,. \tag{9}$$

We can extend this concept to the space of probability distributions (*Wasserstein gradient flow*) [3]. Let us consider the space of probability measures $\mathcal{P}_2(\mathcal{M})$, that is, the set of probability measures with finite second moments defined on the manifold $\mathcal{M}$:

$$\mathcal{P}_2(\mathcal{M}) = \left\{ \varphi : \mathcal{M} \to [0, \infty) \,\middle|\, \int_{\mathcal{M}} \mathrm{d}\varphi = 1, \quad \int_{\mathcal{M}} |x|^2 \varphi(x) \mathrm{d}x < +\infty \right\}.$$

Taking $\Pi(\mu, \nu)$ as the set of joint probability measures with marginals $\mu, \nu$, we can define the Wasserstein metric on the space $\mathcal{P}_2(\mathcal{M})$ as:

$$W_2^2(\mu, \nu) = \inf_{\pi \in \Pi(\mu, \nu)} \int |x - y|^2 \, \mathrm{d}\pi(x, y) \,. \tag{10}$$

Considering the optimization problem of a functional $J : \mathcal{P}_2(\mathcal{M}) \to \mathbb{R}$, such as the KL divergence between the particle approximation in Eq. (8) and the target posterior $\pi(x)$,

$$\inf_{\rho \in \mathcal{P}_2(\mathcal{M})} D_{KL}(\rho, \pi) = \int_{\mathcal{M}} (\log \rho(x) - \log \pi(x)) \rho(x) \, \mathrm{d}x \,,$$

the evolution in time of the measure $\rho$ under the equivalent of the gradient, the Wasserstein gradient flow, is described by the *Liouville equation*[2] [38, 3, 64]:

$$\begin{aligned}
\frac{\partial \rho(x)}{\partial t} &= \nabla \cdot \left( \rho(x) \nabla \frac{\delta}{\delta \rho} D_{KL}(\rho, \pi) \right) \\
&= \nabla \cdot \left( \rho(x) \nabla \big( \log \rho(x) - \log \pi(x) \big) \right),
\end{aligned} \tag{11}$$

---

[1]Together with the initial condition, this is know as the Cauchy problem.
[2]Also referred to as continuity equation.

where $\nabla \frac{\delta}{\delta \rho} D_{KL}(\rho, \pi) =: \nabla_{\mathcal{W}_2} D_{KL}(\rho, \pi)$ is the Wasserstein gradient and the operator $\frac{\delta}{\delta \rho}$ : $\mathcal{P}_2(\mathcal{M}) \to \mathbb{R}$ represents the functional derivative or first variation (see Appendix C for more details). In the particular case of the KL functional, we can recover the Fokker-Planck equation,

$$\frac{\partial \rho(x)}{\partial t} = \nabla \cdot \big(\rho(x)\nabla(\log \rho(x) - \log \pi(x))\big)$$
$$= -\nabla \cdot \big(\rho(x)\nabla \log \pi(x)\big) + \nabla^2 \rho(x),$$

that admits as unique stationary distribution the posterior $\pi(x)$. The deterministic particle dynamics ODE [2] related to Eq. (11), namely mean-field Wasserstein dynamics, is then given by:

$$\frac{dx}{dt} = -\nabla\big(\log \rho(x) - \log \pi(x)\big). \tag{12}$$

Considering a discretization of Eq. (12) for a particle system $\{x\}_{i=1}^n$ and small stepsize $\epsilon_t$, we can rewrite Eq. (12) as:

$$x_i^{t+1} = x_i^t + \epsilon_t\big(\nabla \log \pi(x_i^t) - \nabla \log \rho(x_i^t)\big). \tag{13}$$

Unfortunately, we do not have access to the analytical form of the gradient $\nabla \log \rho$, so an approximation is needed. At this point, it is crucial to observe the similarity between the discretization of the Wasserstein gradient flow in Eq. (13) and the repulsive update in Eq. (3) to notice how, if the kernelized repulsion is an approximation of the gradient of the empirical particle measure, the update rule minimizes the KL divergence between the particle measure and the target posterior. Different sample-based approximations of the gradient that use a kernel function have been recently studied. The simplest one is given by the kernel density estimation (KDE) (details in Appendix E) $\tilde{\rho}_t(x) = \frac{1}{n}\sum_{i=1}^n k(x, x_t^i)$, where $k(\cdot, \cdot) : \mathbb{R}^d \times \mathbb{R}^d \to \mathbb{R}$ and the gradient of its log density is given by [70]:

$$\nabla \log \rho(x_i^t) \approx \frac{\sum_{j=1}^n \nabla_{x_i^t} k(x_i^t, x_j^t)}{\sum_{j=1}^n k(x_i^t, x_j^t)}. \tag{14}$$

Using this approximation in Eq. (13) we obtain:

$$x_i^{t+1} = x_i^t + \epsilon_t\left(\nabla \log \pi(x_i^t) - \frac{\sum_{j=1}^n \nabla_{x_i^t} k(x_i^t, x_j^t)}{\sum_{j=1}^n k(x_i^t, x_j^t)}\right), \tag{15}$$

where, if we substitute the posterior for $\pi$, we obtain an expression for the repulsive force in Eq. (3). This shows that if the repulsive term in Eq. (3) is the normalized sum of the gradients $\mathcal{R} = \left(\sum_{j=1}^n k(\mathbf{w}_i^t, \mathbf{w}_j^t)\right)^{-1} \sum_{j=1}^n \nabla_{\mathbf{w}_i^t} k(\mathbf{w}_i^t, \mathbf{w}_j^t)$, we do not only encourage diversity of the ensemble members and thus avoid collapse, but surprisingly—in the asymptotic limit of $n \to \infty$, where the KDE approximation is exact [66]—also converge to the true Bayesian posterior!

Nevertheless, approximating the gradient of the empirical measure with the KDE can lead to suboptimal performance, as already studied by Li and Turner [46]. They instead introduced a new *Stein gradient estimator* (SGE) that offers better performance, while maintaining the same computational cost. Even more recently, Shi et al. [69] introduced a spectral method for gradient estimation (SSGE), that also allows for a simple estimation on out-of-sample points. These two estimators can be used in Eq. (13), to formulate the following update rules with two alternative repulsive forces. The one using the Stein estimator, that we will call SGE-WGD, is:

$$x_i^{t+1} = x_i^t + \epsilon_t\left(\nabla \log \pi(x_i^t) + \sum_{j=1}^n (K + \eta\mathbb{I})_{ij}^{-1} \sum_{k=1}^n \nabla_{x_i^t} k(x_k^t, x_j^t)\right), \tag{16}$$

where $K$ is the kernel Gram matrix, $\eta$ a small constant, and $\mathbb{I}$ the identity matrix. We can notice an important difference between KDE and SGE, in that the former is only considering the interaction of the $i$-th particle being updated with all the others, while the latter is simultaneously considering also the interactions between the remaining particles. The spectral method, that we will call SSGE-WGD, leads to the following update rule:

$$x_{t+1}^i = x_t^i + \epsilon_t\left(\nabla \log \pi(x_t^i) + \sum_{j=1}^J \frac{1}{\lambda_j^2} \sum_{m=1}^n \sum_{k=1}^n u_{jk}\nabla_{x_m} k(x_m^t, x_k^t) \cdot \sum_{l=1}^n u_{jl} k(x_i^t, x_l^t)\right) \tag{17}$$

where $\lambda_j$ is the $j$-th eigenvalue of the kernel matrix and $u_{jk}$ is the $k$-th component of the $j$-th eigenvector. Computationally, both SSGE and SGE have a cost of $\mathcal{O}(M^3 + M^2 d)$, with $M$ being the number of points and $d$ their dimensionality. SSGE has an additional cost for predictions of $\mathcal{O}(M(d + J))$, where $J$ is the number of eigenvalues.

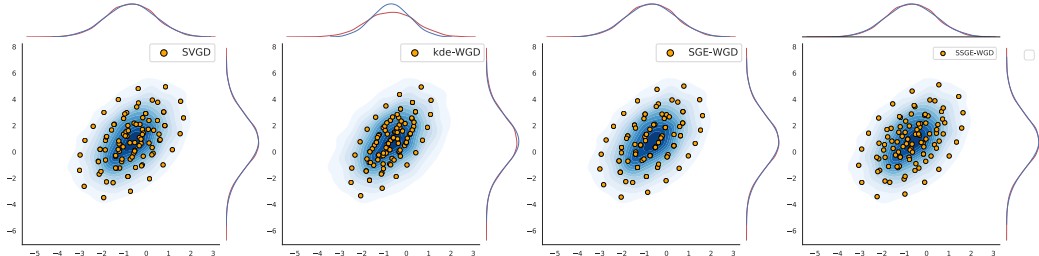

Figure 2: **Single Gaussian.** We show samples from SVGD, KDE-WGD, SGE-WGD, and SSGE-WGD (from left to right). The upper and right plots show the empirical one-dimensional marginal distributions obtained using KDE on the samples (red) and on the particles (blue).

### 3.3 Gradient flow in function space

To theoretically justify the update rule introduced in function space in Eq. (5), we can rewrite the Liouville equation for the gradient flow in Eq. (11) in function space as

$$
\begin{aligned}
\frac{\partial \rho(\boldsymbol{f})}{\partial t} &= \nabla \cdot \left( \rho(\boldsymbol{f}) \nabla \frac{\delta}{\delta \rho} D_{KL}(\rho, \pi) \right) \\
&= \nabla \cdot \left( \rho(\boldsymbol{f}) \nabla \big( \log \rho(\boldsymbol{f}) - \log \pi(\boldsymbol{f}) \big) \right).
\end{aligned}
\tag{18}
$$

Following this update, the mean field functional dynamics are

$$
\frac{d\boldsymbol{f}}{dt} = -\nabla \big( \log \rho(\boldsymbol{f}) - \log \pi(\boldsymbol{f}) \big).
\tag{19}
$$

Using the same KDE approximation as above, we can obtain a discretized evolution in function space and with it an explicit form for the repulsive force in Eq. (4) as

$$
\boldsymbol{f}_{t+1}^i = \boldsymbol{f}_t^i + \epsilon_t \left( \nabla_{\boldsymbol{f}} \log \pi(\boldsymbol{f}_t^i) - \frac{\sum_{j=1}^n \nabla_{\boldsymbol{f}_i^t} k(\boldsymbol{f}_i^t, \boldsymbol{f}_j^t)}{\sum_{j=1}^n k(\boldsymbol{f}_i^t, \boldsymbol{f}_j^t)} \right).
\tag{20}
$$

The update rules using the SGE and SSGE approximations follow as for the parametric case. It is important to notice that this update rule requires the function space prior gradient: $\nabla_{\boldsymbol{f}_j} \log p(\boldsymbol{f}_j|\boldsymbol{x}, \boldsymbol{y}) = \nabla_{\boldsymbol{f}_j} \log p(\boldsymbol{y}|\boldsymbol{x}, \boldsymbol{f}_j) + \nabla_{\boldsymbol{f}_j} \log p(\boldsymbol{f}_j)$. If one wants to use an implicit prior defined in weight space, an additional estimator is needed due to its analytical intractability. We again adopted the SSGE, introduced by Shi et al. [69], which was already used for a similar purpose in Sun et al. [71]. It is also interesting to note that the update rule in Eq. (20) readily allows for the use of alternative priors that have an analytical form, such as Gaussian processes. This is an important feature of our method that allows for an explicit encoding of function space properties that can be useful for example in achieving better out of distribution detection capabilities [14].

### 3.4 The choice of the kernel

A repulsive effect can always be created in the ensemble by means of the gradient of any kernel function that is measuring the similarity between two members. Nevertheless, it is important to keep in mind that to ensure the asymptotic convergence to the Bayesian posterior, the repulsive component must be a consistent estimator of the gradient in Eq. (13), as shown in Section 3. Therefore, some important constraints over the kernel choice are needed. In particular, the SGE and SSGE need a kernel function belonging to the Stein class (see Shi et al. [69] for more details). On the other hand, for the KDE, any symmetric probability density function can be used. On this subject, the work of Aggarwal et al. [1] has shown how the Manhattan distance metric (L1 norm) might be preferable over the Euclidean distance metric (L2 norm) for high-dimensional settings. We performed some additional experiments using the L1 norm (Laplace kernel) for the KDE but we could not observe any substantial difference compared to using the L2 norm. Further investigations regarding this hypothesis are left for future research.

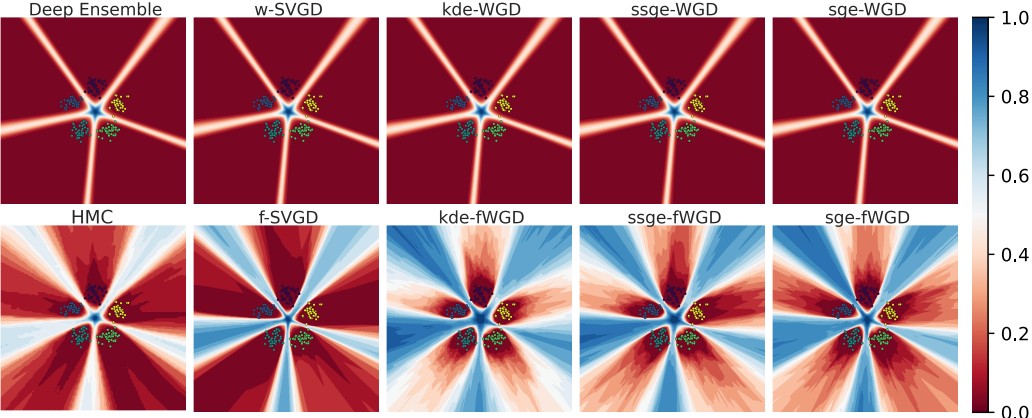

Figure 3: **BNN 2D classification.** We show the entropy of the predictive posteriors. Again, the function-space methods capture the uncertainty better than the weight-space ones, thus approaching the gold-standard HMC posterior.

## 4  Experiments

In this section, we compare the different proposed WGD methods with deep ensembles and SVGD on synthetic sampling, regression, and classification tasks and real-world image classification tasks. We use an RBF kernel (except where otherwise specified) with the popular median heuristic [51] to choose the kernel bandwidth. In our experiments, an adaptive bandwidth leads to better performance than fixing and tuning a single constant value for the entire evolution of the particles. We also quantitatively assess the uncertainty estimation of the methods in terms of calibration and OOD detection. In our experiments, we report the test accuracy, negative log-likelihood (NLL), and the expected calibration error (ECE) [53]. To assert the robustness on out-of-distribution (OOD) data, we report the ratio between predictive entropy on OOD and test data points ($H_o/H_t$), and the OOD detection area under the ROC curve AUROC(H) [47]. Moreover, to assess the diversity of the ensemble generated by the different methods in function space, we measure the functional diversity using the model disagreement (MD) (details in Appendix B). In particular, we report the ratio between the average model disagreement on the OOD and test data points ($MD_o/MD_t$) and additionally the OOD detection AUROC(MD) computed using this measure instead of the entropy.

**Sampling from synthetic distributions**   As a sanity check, we first assessed the ability of our different approximations for Wasserstein gradient descent (using KDE, SGE, and SSGE) to sample from a two-dimensional Gaussian distribution (Figure 2). We see that our SGE-WGD, SSGE-WGD and the SVGD fit the target almost perfectly. We also tested the different methods in a more complex two-dimensional Funnel distribution [59] and present the results in Figure F.1 in the Appendix. There, SGE-WGD and SVGD also perform best.

**BNN 1D regression**   We then assessed the different methods in fitting a BNN posterior on a synthetically generated one-dimensional regression task. The results are reported in Figure 1, consisting of the mean prediction and $\pm 1, 2, 3$ standard deviations of the predictive distribution. We can see that all methods performing inference in the weight space (DE, w-SVGD, WGD) are unable to capture the epistemic uncertainty between the two clusters of training data points. Conversely, the functional methods (f-SVGD, fWGD) are perfectly able to infer the diversity of the hypotheses in this region due to the lack of training evidence. They thereby achieve a predictive posterior that very closely resembles the one obtained with the gold-standard HMC sampling.

**BNN 2D classification**   Next, we investigated the predictive performance and quality of uncertainty estimation of the methods in a two-dimensional synthetic classification setting. The results are displayed in Figure 3. We can clearly observe that the weight-space methods are overconfident and do not capture the uncertainty well. Moreover, all the functions seems to collapse to the optimal classifier. These methods thus only account for uncertainty close to the decision boundaries and to the origin

Table 1: **BNN image classification.** **AUROC(H)** is the AUROC computed using the entropy whereas **AUROC(MD)** is computed using the model disagreement. $\mathbf{H_o/H_t}$ is the ratio of the entropies on OOD and test points respectively and $\mathbf{MD_o/MD_t}$ is the ratio for model disagreement. We see that the best accuracy is achieved by our WGD methods, while our fWGD methods yield the best OOD detection and funtional diversity. All our proposed methods improve over standard deep ensembles in terms of accuracy and diversity, highlighting the effect of our repulsion.

| | | AUROC(H) | AUROC(MD) | Accuracy | $\mathbf{H_o/H_t}$ | $\mathbf{MD_o/MD_t}$ | ECE | NLL |
|---|---|---|---|---|---|---|---|---|
| FashionMNIST | Deep ensemble [42] | 0.958±0.001 | 0.975±0.001 | 91.122±0.013 | 6.257±0.005 | 6.394±0.001 | **0.012±0.001** | 0.129±0.001 |
| | SVGD [51] | 0.960±0.001 | 0.973±0.001 | 91.134±0.024 | 6.315±0.019 | 6.395±0.018 | 0.014±0.001 | 0.127±0.001 |
| | f-SVGD [76] | 0.956±0.001 | 0.975±0.001 | 89.884±0.015 | 5.652±0.009 | 6.531±0.005 | 0.013±0.001 | 0.150±0.001 |
| | hyper-DE [80] | 0.968±0.001 | **0.981±0.001** | 91.160±0.007 | 6.682±0.065 | 7.059±0.152 | 0.014±0.001 | 0.128±0.001 |
| | kde-WGD (ours) | 0.960±0.001 | 0.970±0.001 | 91.238±0.019 | 6.587±0.019 | 6.379±0.018 | 0.014±0.001 | 0.128±0.001 |
| | sge-WGD (ours) | 0.960±0.001 | 0.970±0.001 | **91.312±0.016** | 6.562±0.007 | 6.363±0.009 | **0.012±0.001** | 0.128±0.001 |
| | ssge-WGD (ours) | 0.968±0.001 | 0.979±0.001 | 91.198±0.024 | 6.522±0.009 | 6.610±0.012 | **0.012±0.001** | 0.130±0.001 |
| | kde-fWGD (ours) | **0.971±0.001** | **0.980±0.001** | 91.260±0.011 | 7.079±0.016 | 6.887±0.015 | 0.015±0.001 | **0.125±0.001** |
| | sge-fWGD (ours) | 0.969±0.001 | 0.978±0.001 | 91.192±0.013 | 7.076±0.004 | 6.900±0.005 | 0.015±0.001 | **0.125±0.001** |
| | ssge-fWGD (ours) | **0.971±0.001** | **0.980±0.001** | 91.240±0.022 | **7.129±0.006** | **6.951±0.005** | 0.016±0.001 | **0.124±0.001** |
| CIFAR10 | Deep ensemble [42] | **0.843±0.004** | 0.736±0.005 | 85.552±0.076 | **2.244±0.006** | 1.667±0.008 | 0.049±0.001 | 0.277±0.001 |
| | SVGD [51] | 0.825±0.001 | 0.710±0.002 | 85.142±0.017 | 2.106±0.003 | 1.567±0.004 | 0.052±0.001 | 0.287±0.001 |
| | fSVGD [76] | 0.783±0.001 | 0.712±0.001 | 84.510±0.031 | 1.968±0.004 | 1.624±0.003 | 0.049±0.001 | 0.292±0.001 |
| | hyper-DE [80] | 0.789±0.001 | 0.743±0.001 | 84.743±0.011 | 1.951±0.010 | 1.690±0.015 | 0.046±0.001 | 0.288±0.001 |
| | kde-WGD (ours) | 0.838±0.001 | 0.735±0.004 | **85.904±0.030** | 2.205±0.003 | 1.661±0.008 | 0.053±0.001 | **0.276±0.001** |
| | sge-WGD (ours) | 0.837±0.003 | 0.725±0.004 | 85.792±0.035 | 2.214±0.010 | 1.634±0.004 | 0.051±0.001 | **0.275±0.001** |
| | ssge-WGD (ours) | 0.832±0.003 | 0.731±0.005 | 85.638±0.038 | 2.182±0.015 | 1.655±0.001 | 0.049±0.001 | **0.276±0.001** |
| | kde-fWGD (ours) | 0.791±0.002 | **0.758±0.002** | 84.888±0.030 | 1.970±0.004 | **1.749±0.005** | **0.044±0.001** | 0.282±0.001 |
| | sge-fWGD (ours) | 0.795±0.001 | 0.754±0.002 | 84.766±0.060 | 1.984±0.003 | 1.729±0.002 | 0.047±0.001 | 0.288±0.001 |
| | ssge-fWGD (ours) | 0.792±0.002 | 0.752±0.002 | 84.762±0.034 | 1.970±0.006 | 1.723±0.005 | 0.046±0.001 | 0.286±0.001 |

region, for which the uncertainty is purely aleatoric. In this setting, f-SVGD suffers from similar issues as the weight space methods, being overconfident away from the training data. Conversely, our fWGD methods are confident (low entropy) around the data but not out-of-distribution, thus representing the epistemic uncertainty better. This suggests that the functional diversity captured by this method naturally leads to a *distance-aware* uncertainty estimation [49, 20], a property that translates into confident predictions only in the proximity of the training data, allowing for a principled OOD detection.

**FashionMNIST classification**  Moving on to real-world data, we used an image classification setting using the FashionMNIST dataset [83] for training and the MNIST dataset [43] as an out-of-distribution (OOD) task. The results are reported in Table 1 (top). We can see that all our methods improve upon standard deep ensembles and SVGD, highlighting the effectiveness of our proposed repulsion terms when training neural network ensembles. In particular, the sge-WGD offers the best accuracy, whereas the methods in function space all offer a better OOD detection. This is probably due to the fact that these methods achieve a higher entropy ratio and functional diversity measured via the model disagreement when compared to their weight-space counterparts. Interestingly, they also reach the lowest NLL values. We can also notice how the model disagreement (MD) not only serves its purpose as a metric for the functional heterogeneity of the ensemble but also allows for a better OOD detection in comparison to the entropy. To the best of our knowledge, this insight has not been described before, although it has been used in continual learning [31]. Interestingly, using this metric, the hyper-deep ensemble [80] shows OOD detection performance comparable with our repulsive ensemble in function space.

**CIFAR classification**  Finally, we use a ResNet32 architecture [29] on CIFAR-10 [41] with the SVHN dataset [61] as OOD data. The results are reported in Table 1 (bottom). We can see that in this case, the weight-space methods achieve better performance in accuracy and OOD detection using the entropy than the ones in function space. Nevertheless, all our repulsive ensembles improve functional diversity, accuracy, and OOD detection when compared to standard SVGD, whereas the standard deep ensemble achieves the best OOD detection using the entropy.

## 5   Related Work

The theoretical and empirical properties of SVGD have been well studied [40, 48, 13] and it can also be seen as a Wasserstein gradient flow of the KL divergence in the Stein geometry [16, 50]

(see Appendix D for more details). Interestingly, a gradient flow interpretation is also possible for (stochastic gradient) MCMC-type algorithms [48], which can be unified under a general particle inference framework [10]. Moreover, our Wasserstein gradient descent using the SGE approximation can also be derived using an alternative formulation as a gradient flow with smoothed test functions [48]. A projected version of WGD has been studied in Wang et al. [75], which could also be readily applied in our framework. Besides particle methods, Bayesian neural networks [54, 59] have gained popularity recently [79, 22, 19, 36], using modern MCMC [59, 79, 22, 24, 21] and variational inference techniques [5, 72, 17, 34]. On the other hand, ensemble methods have also been extensively studied [42, 18, 82, 23, 80, 32, 85, 78].Moreover, repulsive interactions between the members have also been studied in Wabartha et al. [74]. Moreover, providing Bayesian interpretations for deep ensembles has been previously attempted through the lenses of stationary SGD distributions [56, 8], ensembles of linear models [57], additional random functions [62, 12, 27], approximate inference [82], Stein variational inference [15], and marginal likelihood lower bounds [53], and ensembles have also been shown to provide good approximations to the true BNN posterior in some settings [36]. Furthermore, variational inference in function space has recently gained attention [71] and the limitations of the KL divergence have been studied in Burt et al. [7].

## 6    Conclusion

We have presented a simple and principled way to improve upon standard deep ensemble methods. To this end, we have shown that the introduction of a kernelized repulsion between members of the ensemble not only improves the accuracy of the predictions but—even more importantly—can be seen as Wasserstein gradient descent on the KL divergence, thus transforming the MAP inference of deep ensembles into proper Bayesian inference. Moreover, we have shown that incorporating functional repulsion between ensemble members can improve the quality of the estimated uncertainties on simple synthetic examples and OOD detection on real-world data and can approach the true Bayesian posterior more closely.

In future work, it will be interesting to study the impact of the Jacobian in the fWGD update and its implications on the Liouville equation in more detail, also compared to other neural network Jacobian methods, such as neural tangent kernels [37] and generalized Gauss-Newton approximations [35]. Moreover, it would be interesting to derive explicit convergence bounds for our proposed method and compare them to the existing bounds for SVGD [40].

### Acknowledgments

VF would like to acknowledge financial support from the Strategic Focus Area "Personalized Health and Related Technologies" of the ETH Domain through the grant #2017-110 and from the Swiss Data Science Center through a PhD fellowship. We thank Florian Wenzel, Alexander Immer, Andrew Gordon Wilson, Pavel Izmailov, Christian Henning, and Johannes von Oswald for helpful discussions. We also thank Dr. Sheldon Cooper, Dr. Leonard Hofstadter and Penny Hofstadter for their support and inspiration.

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
