# Supplementary Material: Repulsive Deep Ensembles are Bayesian

## A    Non-identifiable neural networks

Deep neural networks are parametric models able to learn complex non-linear functions from few training instances and thus can be deployed to solve many tasks. Their overparameterized architecture, characterized by a number of parameters far larger than that of training data points, enables them to retain entire datasets even with random labels [84]. Even more, this overparameterized regime makes neural network approximations of a given function not unique in the sense that multiple configurations of weights might lead to the same function. Indeed, the output of a feed forward neural network given some fixed input remains unchanged under a set of transformations. For instance, certain weight permutations and sign flips in MLPs leave the output unchanged [9]. The invariance of predictions and therefore of parameterized functions under a given weight transformation translates to invariance of the likelihood function. This effect is commonly denoted as *non-identifiability* of neural networks [68]. More in detail, let $g : (\boldsymbol{x}, \mathbf{w}) \mapsto f(\boldsymbol{x}; \mathbf{w})$ be the map that maps a data point $\boldsymbol{x} \in \mathcal{X}$ and a weight vector $\mathbf{w} \in \mathbb{R}^d$ to the corresponding neural network output and denote $\boldsymbol{f}_i := f(\boldsymbol{x}; \mathbf{w}_i)$ the output with a certain configuration of weights $\mathbf{w}_i$. Then for any non identifiable pair $\{\mathbf{w}_i, \mathbf{w}_j\} \in \mathcal{W} \subseteq \mathbb{R}^d$ and $\boldsymbol{f}_i, \boldsymbol{f}_j \in \mathcal{F}$ their respective functions:

$$\boldsymbol{f}_i = \boldsymbol{f}_j \quad \Longrightarrow \quad p(\mathbf{w}_i|\mathcal{D}) = p(\mathbf{w}_j|\mathcal{D}) \quad \not\Longrightarrow \quad \mathbf{w}_i = \mathbf{w}_j \,.$$

Strictly speaking, the map $g : \mathcal{X} \times \mathcal{W} \to \mathcal{F}$ is not injective (many to one). Denoting by $T$ the class of transformations under which a neural network is non-identifiable in the weights space, it is always possible to identify a cone $K \subset \mathbb{R}^d$ such that for any parameter configuration $\mathbf{w}$ there exist a point $\eta \in K$ and a transformation $\tau \in T$ for which it holds that $\tau(\eta) = \mathbf{w}$. This means that every parameter configuration has an equivalent on the subset given by the cone [30]. Modern neural networks containing convolutional and max-pooling layers have even more symmetries than MLPs [4]. Given that in practice we cannot constraint the support of the posterior distribution to be the cone of identifiable parameter configurations and given that the likelihood model is also invariant under those transformations that do not change the function, the posterior landscape includes multiple equally likely modes that despite their different positions represent the same function. It is important to notice that this is always true for the likelihood but not for the posterior. Indeed, for the modes to be equally likely, the prior should also be invariant under those transformations, condition that is in general not true. Nevertheless, the fact that there are multiple modes of the posterior parametrizing for the same function remains true but they might be arbitrarily re-scaled by the prior[3]. As we will see in the following, this redundancy of the posterior is problematic when we want to obtain samples from it. Moreover it is interesting to notice how this issue disappears when the Bayesian inference is considered in the space of functions instead of weights. In this case, indeed, every optimal function has a unique mode in the landscape of the posterior and redundancy is not present:

$$\boldsymbol{f}_i \neq \boldsymbol{f}_j \quad \Longrightarrow \quad p(\boldsymbol{f}_i|\mathcal{D}) \neq p(\boldsymbol{f}_j|\mathcal{D}) \,.$$

In spite of that, performing inference over distributions of functions is prohibitive in practice due to the infinite dimensionality of the space in consideration. Only in very limited cases like the one of Gaussian Proccess, Bayesian inference is exact. Interestingly Neural network model in the limit of infinite width are Gaussian processes with a particular choice of the kernel determined by the architecture [44, 81, 60]. In this limit Bayesian inference over functions can be performed analytically.

## B    Quantify functional diversity

As illustrated in Section 2 , in the Bayesian context, predictions are made by doing a Monte-Carlo estimation of the BMA. Functional diversity, and so the diversity in the hypotheses taken in consideration when performing the estimation, determines the epistemic uncertainty and the confidence over the predictions. Importantly, the epistemic uncertainty allows for the quantification

---

[3]Note that for the fully factorized Gaussian prior commonly adopted, the invariance under permutations is true

of the likelihood of a test point to belong to the same distribution from which the training data points were sampled [65]. Following this, the uncertainty can be used for the problem of Out-of-distribution (OOD) detection [11] that is often linked to the ability of a model to *"know what it doesn't know"*. A common way used in the literature to quantify the uncertainty is the Entropy[4] $\mathcal{H}$ of the predictive distribution:

$$\mathcal{H}\{p(\mathbf{y}'|\mathbf{x}', \mathcal{D})\} = -\sum_y p(\mathbf{y}'|\mathbf{x}', \mathcal{D}) \log p(\mathbf{y}'|\mathbf{x}', \mathcal{D}). \tag{21}$$

Nevertheless, it has been argued in recent works [55] that this is not a good measure of uncertainty because it does not allow for a disentanglement of epistemic and aletoric uncertainty. Intuitively, we would like the predictive distribution of an OOD point to be uniform over the different classes. However, using the entropy and so the average prediction in the BMA, we are not able to distinguish between the case in which all the hypotheses disagree very confidently due to the epistemic uncertainty or are equally not confident due to the aleatoric uncertainty. To overcome this limitation, we can use a direct measure of the model disagreement computed as:

$$\mathcal{MD}^2(\mathbf{y}'; \mathbf{x}', \mathcal{D}) = \int_{\mathbf{w}} \left[ p(\mathbf{y}'|\mathbf{x}', \mathbf{w}) - p(\mathbf{y}'|\mathbf{x}', \mathcal{D}) \right]^2 p(\mathbf{w}|\mathcal{D}) d\mathbf{w}. \tag{22}$$

It is easy to see how the quantity in Eq. (22), measuring the deviation from the average prediction is zero when all models agree on the prediction. The latter can be the case of a training point where all hypotheses are confident or a noisy point where all models *"don't know"* the class and are equally uncertain. On the other side the model disagreement will be greater the zero the more the model disagree on a prediction representing like this the epistemic uncertainty. To obtain a scalar quantity out of Eq. (22) we can consider the expectation over the output space of $\mathbf{y}$:

$$\mathcal{MD}^2(\mathbf{x}') = \mathbb{E}_y \left[ \int_{\mathbf{w}} \left[ p(\mathbf{y}'|\mathbf{x}', \mathbf{w}) - p(\mathbf{y}'|\mathbf{x}', \mathcal{D}) \right]^2 p(\mathbf{w}|\mathcal{D}) d\mathbf{w} \right]. \tag{23}$$

## C Functional derivative of the KL divergence

In this section, we show the derivation of the functional derivative for the KL divergence functional. We start with some preliminary definitions.

Given a manifold $\mathcal{M}$ embedded in $\mathbb{R}^d$, let $F[\rho]$ be a functional, i.e. a mapping from a normed linear space of function (Banach space) $\mathcal{F} = \{\rho(x) : x \in \mathcal{M}\}$ to the field of real numbers $F : \mathcal{F} \to \mathbb{R}$. The functional derivative $\delta F[\rho]/\delta\rho(x)$ represents the variation of value of the functional if the function $\rho(x)$ is changed.

**Definition 2** (Functional derivative). *Given a manifold $\mathcal{M}$ and a functional $F : \mathcal{F} \to \mathbb{R}$ with respect to $\rho$ is defined as:*

$$\int \frac{\delta F}{\delta\rho(x)} \phi(x) dx = \lim_{\epsilon \to 0} \frac{F[\rho(x) + \epsilon\phi(x)] - F(\rho(x))}{\epsilon} = \frac{d}{d\epsilon} F[\rho(x) + \epsilon\phi(x)] \Big|_{\epsilon=0} \tag{24}$$

*for every smooth $\phi$.*

**Definition 3** (KL divergence). *Given $\rho$ and $\pi$ two probability densities on $\mathcal{M}$, the KL-divergence is defined as:*

$$D_{KL}(\rho, \pi) = \int_{\mathcal{M}} \left( \log\rho(x) - \log\pi(x) \right) \rho(x) \, dx. \tag{25}$$

**Proposition 1.** *The functional derivative of the KL divergence in Eq. (25) is:*

$$\frac{\delta D_{KL}}{\delta\rho(x)} = \log\frac{\rho(x)}{\pi(x)} + 1 \tag{26}$$

---

[4]The continuous case is analogous using the differential entropy

*Proof.* using the definition of functional derivative in Eq. (24) :

$$
\begin{aligned}
\int \frac{\delta D_{KL}}{\delta \rho(x)} \phi(x) dx &= \frac{d}{d\epsilon} D_{KL}(\rho + \epsilon \phi, \pi) \Big|_{\epsilon=0} \\
&= \int \frac{d}{d\epsilon} \left[ (\rho(x) + \epsilon \phi(x)) \log \frac{(\rho(x) + \epsilon \phi(x))}{\pi(x)} \right]_{\epsilon=0} dx \\
&= \int \left[ \phi(x) \log \frac{(\rho(x) + \epsilon \phi(x))}{\pi(x)} + \frac{d(\rho(x) + \epsilon \phi(x))}{d\epsilon} \right]_{\epsilon=0} dx \\
&= \int \left[ \log \frac{\rho(x)}{\pi(x)} + 1 \right] \phi(x) dx
\end{aligned}
\tag{27}
$$

$\square$

# D  SVGD as Wasserstein gradient flow

To understand the connection between the Wasserstein gradient flow and the SVGD method, we first need to introduce the concept of gradient flow on a manifold. Let's consider a Riemannian manifold $\mathcal{M}$ equipped with the metric tensor $G(x)$ defined for all $x \in \mathcal{M}$. Here, $G(x) : \mathcal{T}_{\mathcal{W}} \times \mathcal{T}_{\mathcal{W}} \to \mathbb{R}$ defines a smoothly varying local inner product on the tangent space at each point of the manifold $x$. For manifolds over $\mathbb{R}^d$, the metric tensor is a positive definite matrix that defines local distances for infinitesimal displacements $d(x, x + dx) = \sqrt{dx^\top G(x) dx}$. Considering a functional $J : \mathcal{M} \to \mathbb{R}$, the evolution in Eq. (9) and so the gradient flow, becomes:

$$
\frac{dx}{dt} = -G(x)^{-1} \nabla J(x) .
\tag{28}
$$

We see that the metric tensor of the manifold acts like a perturbation of the gradient. Secondly, we need to reformulate the update equation in 6 in continuous time, as the following ODE:

$$
\frac{dx_i}{dt} = \frac{1}{n} \sum_{j=1}^{n} [k(x_j, x_i) \nabla_{x_j} \log p(x_j) + \nabla_{x_j} k(x_j, x_i)]
\tag{29}
$$

that in the mean field limit $n \to \infty$ becomes:

$$
\begin{aligned}
\frac{dx}{dt} &= \int \left[ k(x', x) \nabla_{x'} \log \pi(x') + \nabla_{x'} k(x', x) \right] \rho(x') dx' \\
&= \int k(x', x) \nabla_{x'} \log \pi(x') \rho(x') dx' + \int \nabla_{x'} k(x', x) \rho(x') dx' .
\end{aligned}
\tag{30}
$$

Due to the boundary condition of a Kernel in the Stein class (see Liu et al. [52] for more details), without loss of generality, we can rewrite the integrals in the previous equation as:

$$
\frac{dx}{dt} = \int k(x', x) \nabla_{x'} \log \pi(x') \rho(x') dx' + \int \nabla_{x'} k(x', x) \rho(x') dx' - \underbrace{k(x', x) \rho(x') \Big|_{||x'|| \to \infty}}_{=0}
$$

and notice how the second and the third terms are the result of an integration by parts:

$$
\begin{aligned}
\frac{dx}{dt} &= \int k(x', x) \nabla_{x'} \log \pi(x') \rho(x') dx' - \int \nabla_{x'} k(x', x) \nabla_{x'} \rho(x') dx' \\
&= \int k(x', x) \nabla_{x'} \log \pi(x') \rho(x') dx' - \int \nabla_{x'} k(x', x) \nabla_{x'} \log \rho(x') \rho(x') dx' \\
&= \int k(x', x) \nabla_{x'} \left[ \log \pi(x') - \log \rho(x') \right] \rho(x') dx' \\
&= \mathbb{E}_{x' \sim \rho} \left[ k(x', x) \nabla_{x'} (\log \pi(x') - \log \rho(x')) \right]
\end{aligned}
\tag{31}
$$

which is exactly the functional derivative of the KL divergence in 11 approximated in the RKHS of the kernel [16, 50]. Following this, the Liouville equation of the SVGD dynamics in the mean field

limit is:

$$\frac{\partial \rho(x)}{\partial t} = -\nabla \cdot \left( \int k(x', x) \nabla_{x'} \big[ \log \pi(x') - \log \rho(x') \big] \rho(x') dx' \right)$$

$$= \nabla \cdot \left( \int k(x', x) \nabla_{x'} \frac{\delta}{\delta \rho} D_{KL}(\rho, \pi) \rho(x') dx' \right). \tag{32}$$

Defining the linear operator $(\mathcal{K}_\rho \phi)(x) := \mathbb{E}_{x' \sim \rho}[k(x', x)\phi(x')]$ the ODE in Eq. (31) becomes:

$$\frac{dx}{dt} = \mathcal{K}_\rho \nabla_{x'} \big( \log \pi(x') - \log \rho(x') \big). \tag{33}$$

And the Liouville equation describing the evolution of the empirical measure of the particles:

$$\frac{\partial \rho}{\partial t} = \nabla \cdot \big( \rho(x)\mathcal{K}_\rho \nabla_{x'} (\log \rho(x') - \log \pi(x')) \big)$$

$$= \nabla \cdot \left( \rho(x)\mathcal{K}_\rho \nabla_{x'} \frac{\delta}{\delta \rho} D_{KL}(\rho, \pi) \right). \tag{34}$$

Notice how the only solution of the previous equation is $\rho = \pi$. If we now compare Eq. (34) and Eq. (11) we can see how they only differ for the application of the operator $\mathcal{K}_\rho$; moreover the operator seems to act like a perturbation of the gradient and defines a local geometry in the same way the Riemannian metric tensor $G(x)$ is doing in 28. The intuition suggest then that the SVGD can be reinterpreted as a gradient flow in the Wasserstein space under a particular *Stein geometry* [16] induced on the manifold by the kernel and where $\mathcal{K}_\rho$ is the metric.

# E    Kernel density estimation

Kernel Density Estimation (KDE) is a nonparametric density estimation technique [66]. When an RBF kernel is used, it can be thought as a smoothed version of the empirical data distribution. Given some training datapoints $\mathcal{D} = \{\mathbf{x}_1, ..., \mathbf{x}_n\}$ with $\mathbf{x}_i \sim p(\mathbf{x})$ and $\mathbf{x} \in \mathbb{R}^D$ their empirical distribution $q_0(\mathbf{x})$ is a mixture of $n$ Dirac deltas centered at each training data:

$$q_0(\mathbf{x}) = \frac{1}{N} \sum_{i=1}^{N} \delta(\mathbf{x} - \mathbf{x}_i). \tag{35}$$

We can now smooth the latter by replacing each delta with an RBF kernel:

$$k_\epsilon(\mathbf{x}, \mathbf{x}_i) = \frac{1}{h} \exp \left( -\frac{||\mathbf{x} - \mathbf{x}_i||^2}{h} \right) \tag{36}$$

where $h > 0$. The kernel density estimator is then defined as:

$$q_h(\mathbf{x}) = \frac{1}{N} \sum_{i=1}^{N} k_h(\mathbf{x}, \mathbf{x}_i) \tag{37}$$

In the limit of $h \to 0$ and $N \to \infty$ the kernel density estimator is unbiased: it is equal to the true density. Indeed $k_{h \to 0}(\mathbf{x}, \mathbf{x}_i) \to \delta(\mathbf{x} - \mathbf{x}_i)$ and so $q_{h \to 0}(\mathbf{x}) \to q_0(\mathbf{x})$ and:

$$\lim_{N \to \infty} q_0(\mathbf{x}) = \lim_{N \to \infty} \frac{1}{N} \sum_{i=1}^{N} (\delta(\mathbf{x} - \mathbf{x}_i))$$

$$= \mathbb{E}_{p(\mathbf{x}')} [\delta(\mathbf{x} - \mathbf{x}')]$$

$$= \int_{\mathbb{R}^D} \delta(\mathbf{x} - \mathbf{x}')p(\mathbf{x}') \, d\mathbf{x}' = p(\mathbf{x}) \tag{38}$$

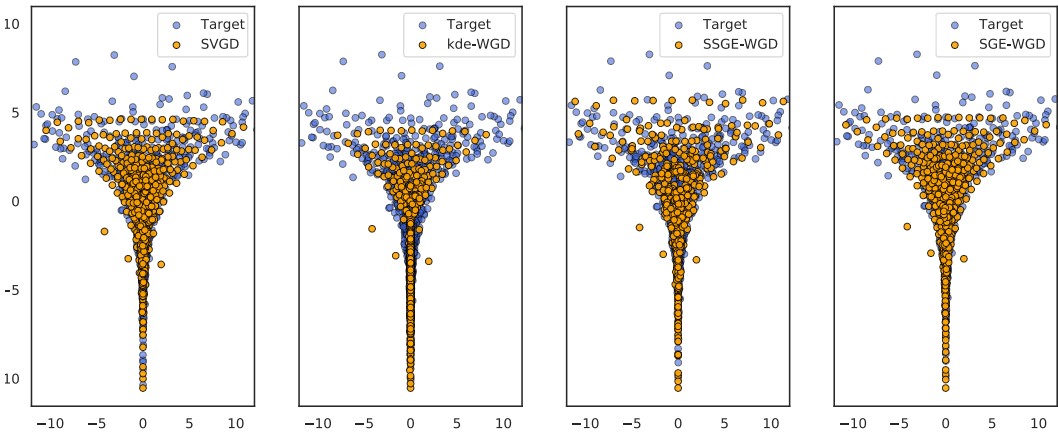

Figure F.1: **Neal's funnel.** The SGE-WGD and SVGD again fit the distribution best.

# F Additional experiments

In this section, we report the additional results for the different methods when sampling from the Funnel distribution $p(x, y) = \mathcal{N}(y|\mu = 0, \sigma = 3)\mathcal{N}(x|0, \exp(y/2))$, the results are reported in Figure F.1.

# G Implementation details

In this section, we report details on our implementation in the experiments we performed. The code is partially based on von Oswald et al. [73] and can be found at https://github.com/ratschlab/repulsive_ensembles. All the experiments were performed on the ETH Leonhard scientific compute cluster with NVIDIA GTX 1080 Ti and took roughly 150 GPU hours.

## G.1 Sampling from synthetic distributions

**Single Gaussian:** we created a two-dimensional Gaussian distribution with mean $\mu = (-0.6871, 0.8010)$ and covariance $\Sigma = \begin{pmatrix} 1.130 & 0.826 \\ 0.826 & 3.389 \end{pmatrix}$. We used a normal initialization with zero mean and standard deviation $\sigma^2 = 3$. We sampled 100 initial particles and optimized them for 5000 iterations using Adam with a fixed learning rate of 0.1. The kernel bandwidth was estimated using the median heuristic for all methods. For the SSGE we used all the eigenvalues. The random seed was fixed to 42.

**Funnel:** the target distribution followed the density $p(x, y) = \mathcal{N}(y|\mu = 0, \sigma = 3)\mathcal{N}(x|0, \exp(y/2))$. We used a normal initialization with zero mean and standard deviation $\sigma^2 = 3$. We sampled 500 initial particles and optimized them for 2000 iterations using Adam with a fixed learning rate of 0.1. The kernel bandwidth was fixed to 0.5 for all methods. For the SSGE we used all the eigenvalues. The random seed was fixed to 42.

## G.2 1D regression

We generated the training data by sampling 45 points from $x_i \sim \text{Uniform}(1.5, 2.5)$ and 45 from $x_i \sim \text{Uniform}(4.5, 6.0)$. The output $y_i$ for a given $x_i$ is then modeled following $y_i = x_i \sin(x_i) + \epsilon_i$ with $\epsilon_i \sim \mathcal{N}(0, 0.25)$. We use a standard Gaussian likelihood and standard normal prior $\mathcal{N}(0, \mathbb{I})$. The model is a feed-forward neural network with 2 hidden layers and 50 hidden units with ReLU activation function. We use 50 particles initialized with random samples from the prior and optimize them using Adam [39] with 15000 gradient steps, a learning rate of 0.01 and batchsize 64. The kernel

bandwidth is estimated using the median heuristic. We tested the models on 100 uniformly distributed points in the interval $[0, 7]$. The random seed was fixed to 42.

### G.3    2D classification

We generate 200 training data points sampled from a mixture of 5 Gaussians with means equidistant on a ring of radius 5 and unitary covariance. The model is a feed-forward neural network with 2 hidden layers and 50 hidden units with ReLU activation function. We use a softmax likelihood and standard normal prior $\mathcal{N}(0, \mathbb{I})$. We use 100 particles initialized with random samples from the prior and optimize them using Adam [39] with 10,000 gradient steps, a learning rate of 0.001 and batchsize 64. The kernel bandwidth is estimated using the median heuristic. The random seed was fixed to 42.

### G.4    Classification on FashionMNIST

On this dataset, we use a feed-forward neural network with 3 hidden layers and 100 hidden units with ReLU activation function. We use a softmax likelihood and standard normal prior $\mathcal{N}(0, \mathbb{I})$. We use 50 particles initialized with random samples from the prior and optimize them using Adam [39] for 50000 steps, a learning rate was 0.001 for sge-WGD,kde-WG,ssge-WGD and 0.0025 for kde-fWGD,ssge-fWGD,sge-fWGD, Deep ensemble, fSVGD, SVGD, and batchsize was 256. The kernel bandwidth is estimated using the median heuristic for all different methods. The learning rates were searched over the following values $(1e-4, 5e-4, 1e-3, 5e-3, 25e-4)$ we tested for 50000 and 30000 total number of iterations, 50 and 100 particles and batchsize 256 and 128. For the hyper-deep ensemble, we implemented it on the L2 parameter by first running a random search to select a set of 5 top values in the range $[1e-3, 1e3]$, which we subsequently used to create an ensemble with the same number of members as the other methods. All results in Table 1 are averaged over the following random seeds $(38, 39, 40, 41, 42)$.

### G.5    Classification on CIFAR-10

On this dataset, we used a residual network (ResNet32) with ReLU activation function. We use a softmax likelihood and standard normal prior $\mathcal{N}(0, 0.1\mathbb{I})$. We use 20 particles initialized using He initialization [28] and optimize them using Adam [39] for 50000 steps, a learning rate was 0.00025 for sge-fWGD,kde-fWGD,ssge-fWGD,fSVGD and 0.0005 for kde-WGD,ssge-WGD,sge-fWGD, Deep ensemble and SVGD, and batchsize was 128. The kernel bandwidth is estimated using the median heuristic for all different methods. The learning rates were searched over the following values $(1e-4, 5e-4, 1e-3, 5e-3, 25e-4, 5e-5)$ we tested for 50000 and 30000 total number of iterations, 20 and 10 particles. For the hyper-deep ensemble, we implemented it on the L2 parameter by first running a random search to select a set of 5 top values in the range $[1e-3, 1e3]$, which we subsequently used to create an ensemble with the same number of members as the other methods. All results in Table 1 are averaged over the following random seeds $(38, 39, 40, 41, 42)$.