# OpenReview forum: "Repulsive Deep Ensembles are Bayesian"
_NeurIPS.cc/2021/Conference — NeurIPS 2021 Spotlight_

### Official Review · Reviewer_M3i3 · 2021-07-14

**Rating:** 8
**Confidence:** 3

**Summary:**

This paper introduces a novel but simple repulsive term to the training objective for deep ensembles which prevents the ensemble members from collapsing to the same point in *function* space. This training procedure is closely related to that of Stein variational gradient descent. Furthermore, the authors show that this repulsive term turns deep ensemble training into approximate Bayesian inference. Finally, the authors demonstrate that this method improves on both SVGD and deep ensembles in a range of benchmark tasks.

**Ethical Concerns:**

None.

**Limitations And Societal Impact:**

The authors have not gone into much detail about the limitations of their method. As mentioned above it would be great to have a discussion of:
1. Memory requirements for training repulsive ensembles.
2. The importance of the choice of kernel and the general importance of hyperparameter tuning.


**Main Review:**

Overall, I found this to be an interesting and thought-provoking paper which I am glad to have had the chance to review. I believe that this is an important area of research and that this paper could lead to interesting future work (or be otherwise of significance). As far as I can tell the proposed method is sound and well-motivated from multiple perspectives.

However, while the experimental evaluation is decent, I think there is definitely room for improvement, which I will describe in more detail below. Furthermore, I think that the biggest weakness of the paper is in clarity. I found section 3, and to a lesser extent section 2, difficult to follow. I'll elaborate on the clarity issues below. I also think that the authors should add a discussion of the limitations of their method. For example, it seems to me that one potential issue in this work is that it requires that all of the ensemble members fit in GPU memory in order to do the training efficiently, but perhaps this isn't the case? Similarly, it also seems that the choice of kernel function must matter a lot for the performance of the method, but this isn't discussed. In particular, I am wondering whether a poor choice might lead to worse performance if the repulsive term dominates the training objective (i.e. how much tuning was required to get training to work)?

### Experimental suggestions

The major weakness of the experimental evaluation was the omission of hyper-deep ensembles as a baseline in at least the MNIST and CIFAR10 experiments. The authors do cite this method, but I think that, as a simple to implement and highly performant baseline which also relies on the increased diversity of ensemble members, this is a must-have comparison.

The other missing ingredient in the experimental sections were some studies to investigate/confirm why/how the proposed method works. For example, it would be interesting to confirm that flipping the sign of the repulsive term does degrade performance as one would expect. It would also be very interesting to see how the LL and repulsive terms evolve during training for a few different datasets. For example, one might imagine that the repulsive term is larger relative to the LL term as training progresses because the ensemble elements might converge to similar functions.



#### Minor suggestions

1. In the toy regression plots (fig 1), please show some of the individual functions so that one case see how different the fits are for each ensemble member.

2. I would also like to see the same toy regression experiments but with a larger more flexible network i.e. a width of 100 rather than 50 and a depth of 3 instead of 2. In my experience standard deep ensembles of this size are able to capture "in-between" uncertainty, so it would be interesting to see what the repulsive term does in this case.

3. In fig 2, it would be great to have a colour bar to give more context for the uncertainty values. It would also be great to have a larger range so that the behaviour as we ger further away from the training date can be understood better. In particular, it seems like the function space methods will behave nicely but it isn't quite clear given the current range of the plots.

For the CIFAR10 experiments, it would be interesting to see results using CIFAR100 as the OOD dataset. This is much more challenging than SVHN and might be a case where the functional methods can shine given their potential "distance-aware" behaviour.

4. I would also like to see rejection-classification experiments (see "Uncertainty Estimation Using a Single Deep Deterministic Neural Network" by van Amersfoort et al. for examples).

5. I think table 1 should be split into two. One which compares the various methods for WGD, and one which compares the best WGD method for each dataset to the baselines. Comparing all of the WGD methods to the baselines is a slightly unfair comparison.

### Clarity Issues

Unfortunately, I can't give as many concrete suggestions for how to improve sections 2 and 3. However, at a high level, I would say that while the technical details provided are important, perhaps some of the detail should be moved to the appendix. The space in the main text could be used to help the reader by providing further explanation, discussion, and intuition building.

Here is a list of specific (but individually minor) difficulties I had when readings these sections, I hope they provide some insight into why I struggled:

1. The symbols $\mathcal{R}$,  $\mathcal{F}$, and $\mathcal{X}$, were never explicitly introduced and their meaning had to be inferred.

2. In section 3.2, there is a switch from Euclidian space to a Riemannian manifold but it isn't explained why this is done. Some intuition would be very useful.

3. The metric tensor in 3.2 needs more explanation. What is this and why has it been introduced. Some intuition would also be great here.

4. In the equation after line 173 it isn't clear if $rho$ is the same as in Eq. 8.

5. The definition of the Wasserstein metric is confusing and ultimately unnecessary since it is the same as one would find from a quick Google search. I think it should be removed (or moved to an appendix) to improve the flow of the section.

6. On line 176 it isn't clear why we have an "equivalent of the gradient" and not simply a gradient.

7. Equation 12 seems to come out of nowhere. Some intuition would be great.

8. Comparing equations 14 and 3 is difficult when they are on different pages, so it would be helpful if the text explicitly compared the repulsive terms. Also, the sentence from line 186 to 189 is too long and should be split into two.

9. Eqn 16 would be more clear if the substitution of the posterior was already made.

10. Is $\beta$ on line 196 simply the denominator of the fraction in Eqn 16?

11. The fact that the true Bayesian posterior is only recovered in the limit as $n \to \infty$ should be mentioned much earlier in the text.


### Summary

While I enjoyed this paper and hope to see it published, I am unable to give a higher score due to the clarity and experimental issues described above. However, if the authors can make section 3 more approachable, and address my two main experimental concerns, I would be happy to increase my score from 5 to 7 or 8.


### Post-author response

The authors have addressed the majority of my concerns, have provided additional experimental results (and committed to adding further results in the camera ready), and have also acknowledge and proposed solutions to the issues of clarity. Thus, I am raising my score to 8.


**Time Spent Reviewing:**

9

---

> ### Author Response · Authors · 2021-08-10
> **Response to reviewer M3i3**
>
> Thank you very much for your time and feedback! We are delighted to see that you enjoyed reading our paper and are prepared to increase your score when your concerns are addressed. We hope to have fully addressed them in this response and we strongly believe they helped improve the quality of our work. We respond to your different points below.
>
> **Computational considerations:** You are right in observing that, in order for our theoretical guarantees to hold, the particles need to be updated simultaneously. That is, each particle needs to be updated once before the next step of updates can be computed. Note that the particles need not necessarily be in memory at the same time for that. One could imagine a setting where the particles are stored and sequentially loaded in and out of the main memory to compute their respective updates. However, this would probably lead to some memory access overhead in terms of computational time.
> That being said, it is an interesting direction for future work to study whether one could train the particles completely sequentially, that is, train each particle to convergence while repelling it from the already training particles. This would probably lead to a technique similar to universal boosting [1]. Following this direction it would also be interesting to see if the methods could benefit from repulsion with past samples,  intuitively this is going to increase the number of samples used in the different gradient estimators and might reduce their variance.
>
> **Choice of the kernel:** We will clarify that in the experiments, we always use the RBF kernel with the popular median heuristic [2] to choose the kernel bandwidth. In our tests, an adaptive bandwidth leads to better performance than fixing and tuning a single constant value for the entire evolution of the particles. We will mention this in our discussion. We agree that it could be interesting future work to explore different kernels; in particular, following the arguments in [4] where they show that the Manhattan distance metric (L1 norm) is consistently more preferable than the Euclidean distance metric (L2 norm) for high dimensional settings we performed some new experiments using the L1 norm (Laplace kernel) for the KDE and we will add them together with a discussion. Nevertheless, it is important to keep in mind that for the repulsive component to be a consistent gradient estimator some important constraints over the kernel choice are needed. In particular, the SGE and SSGE need a kernel function belonging to the Stein class (see [3] for more details). On the other side for the KDE any symmetrical pdf can be used. In general, we believe that as long as the kernel leads to a consistent gradient estimator the repulsive term should not negatively impact the performance of the method given that in most cases the gradients seem to be underestimated leading to solutions that are closer to the MAP and so to standard deep ensembles. We would like to underline that using the median heuristic not much tuning is necessary given that the bandwidth is computed at each iteration based on the particles and so no kernel parameters need to be manually tuned for the method to work.
>
> **Experiments:** Thank you for suggesting hyper-deep ensembles as a baseline. We agree that they are a natural competitor to our method. We implemented hyper deep ensembles on the L2 parameter by first running a random search to select a set of 5 top values in the range  $[10^{−3}, 10^3 ]$ which we subsequently used to create an ensemble with the same number of members as our methods.
> The results for fashion MNIST are reported in the following table:
>
> |f-MNIST | AUROC(H) | AUROC(MD) | Accuracy | $H_o/H_t$ | $MD_o/MD_t$ | ECE | NLL|
> |------------------|--------------- |-------------- |-------------------- |--------------------- |---------------------- | ---------------------- | ---------------------- |
> | Hyper-DE |0.968±0.001 | 0.981±0.001| 91.160±0.007| 6.682±0.065| 7.059±0.152| 0.014±0.001| 0.128±0.001|
>
> Interestingly on this dataset, the hyper-deep ensemble exhibits ood detection performance comparable with our methods and better accuracy than standard deep ensemble. We will add these results to our analysis. On CIFAR-10 instead, we were not able to improve over standard deep ensemble with a fixed value of the L2 parameter as illustrated in the following table.
>
> | CIFAR-10 | AUROC(H) | AUROC(MD) | Accuracy | $H_o/H_t$ | $MD_o/MD_t$ | ECE | NLL|
> |------------------|--------------- |-------------- |-------------------- |--------------------- |---------------------- | ---------------------- | ---------------------- |
> | Hyper-DE | 0.789±0.001| 0.743±0.001 | 84.743±0.011 | 1.951±0.010 | 1.690±0.015 | 0.046±0.001 | 0.288±0.001 |
>
> This issue might be due to the time constraint that didn't allow us to extensively test this additional baseline. We will therefore perform additional tests to better validate the obtained results.
>
> We also implemented your suggested ablation study of flipping the sign of the repulsive term. Consistently, if the sign is flipped in the sampling experiments reported in figure 2, all the particles are collapsing to the MAP estimate given that the force is now attracting the particles to each other. Also in the other experiments, the same procedure decreases the performance of our method, leading to results that are very similar to the standard deep ensemble as expected.
>
> We also like the idea of plotting the magnitude of the different terms over the course of training and will add those figures to the paper. This allows for a better understanding of the function space and weight space training dynamics.
>
> We will also add single function samples to Fig. 1 to better illustrate the single solutions. Re-run the toy regression with a larger network doesn’t seem to give additional insights indeed in this one-dimensional regression setting and using a standard N(0,1) prior, the variance of the predictive posterior seems to explode for all methods.
>
> We will improve the clarity of the 2d classification plots by adding a colour bar for the uncertainty values and extending the predictions on a larger grid so that the uncertainty capabilities can be better qualitatively assessed.
>
> We will also add OOD experiments with CIFAR-100 and rejection curves to better analyze the full potential of our method.
>
> **Clarity:** We regret that you found parts of the paper hard to follow. We agree that the method section might have been a bit too entangled and we are willing to reformulate it in order to improve clarity. In particular, we will take the following actions regarding your suggestions:
> 1. We will clearly define the different symbols $\mathcal{R}$,$\mathcal{F}$ and $\mathcal{X}$.
> 2. We agree that switching from Euclidean to Riemannian manifold in the gradient flow section is not directly used in the paper. We wanted to include this for completeness and as a mathematical tool to further elaborate on the impact of the jacobian on the gradient flow in function space. Due to the lack of space and time, we left this idea as a future direction and only mentioned it in the conclusion. We agree that it would be beneficial to move that part in the appendix to improve the flow of section 3.2 and mention some additional details regarding how to use the gradient flow on a Riemannian manifold to study our method.
> 3. The same considerations in the previous point apply to the metric tensor introduced in 3.2. We will better illustrate this and move it to the appendix.
> 4. The rho after line 173 is used to generally define the space of interest of all probability measures with certain properties like finite second moments. The one used in equation 8 belongs to this space but it is not the only one. For better clarity, we will use a different symbol there.
> 5. We agree that the Wasserstein metric is not directly necessary to understand the section and our theory. Our intention was to completely define the space of interest and underlying the differences with the euclidean space. Nevertheless, we can refer to the appendix for these additional details.
> 6. On line 176 if we would be taking the evolution under the gradient of the KL only, we would be writing it inside equation 12 as $\nabla D_{KL}(\rho, \pi)$ we instead use the term “equivalent” to underline that the gradient here is the Wasserstein gradient defined in line 178 using the first variation. We will rewrite it more clearly avoiding the confusion deriving from the use of “equivalent”.
> 7. We will add additional details and references to motivate the Liouville equation in 12.
> 8. We will try to reformulate the sections in order to make the comparison between equations 14 and 3 more comfortable.
> 9. We will use the same symbol for the posterior in order to make the connection more immediate.
> 10. Yes, $\beta$ is simply the denominator, we will remove it to avoid misunderstandings.
> 11. We will mention the asymptotic convergence of our method earlier in the text.
>
>
>
> [1] Campbell, T., & Li, X. (2019). Universal boosting variational inference. Advances in Neural Information Processing Systems, 32, 3484-3495.
>
> [2] Liu, Q., & Wang, D. (2016). Stein variational gradient descent: A general purpose bayesian inference algorithm. arXiv preprint arXiv:1608.04471.
>
> [3] Jiaxin Shi, Shengyang Sun, and Jun Zhu. A spectral approach to gradient estimation for implicit distributions. In International Conference on Machine Learning, pages 4644–4653. PMLR, 2018.
>
> [4] Aggarwal, Charu C., Alexander Hinneburg, and Daniel A. Keim. "On the surprising behavior of distance metrics in high dimensional space." International conference on database theory. Springer, Berlin, Heidelberg, 2001.

---

> > ### Comment · Reviewer_M3i3 · 2021-08-19
> > **Thanks for the comprehensive response!**
> >
> > Thanks for responding to my points in such detail and for even including additional results. With all of the additional experimental results, and the proposed clarifications and additions to the text, I am happy to increase my score to 8.

---

### Official Review · Reviewer_QY5N · 2021-07-18

**Rating:** 7
**Confidence:** 3

**Summary:**

Maintaining functional diversity between ensemble members are a large, and still not completely solved problem. The authors are introducing a kernelized repulsive term to the update rule of an ensemble. They show empirically that this increases the functional diversity between ensemble members. The authors also demonstrate that their improvement translates into a proper Bayesian inference.

**Limitations And Societal Impact:**

Yes

**Main Review:**

I like the paper. It addresses an important and still unsolved problem of maintaining functional diversity among ensemble members.  I don't have much to critique. I think this is a worthwhile contribution and that it should get published.

QUESTIONS AND SUGGESTIONS:

1) Do we need to train the full ensemble simultaneously? It seems to me that we do. What are the requirements for how simultaneous exactly does the training have to be in order for this method to work?

2) The models you are looking at on CIFAR-10 in Table 1 are surprisingly week, reaching only mid 80s % of test accuracy. It might be a good idea to show that your technique actually works on SOTA-like models. I can imagine that the effect you observe could be a consequences of your models not being trained to their full potential. A CIFAR-100 verification might also be worth adding, given that CIFAR-10 is a relatively simple and small task.

UPDATE AFTER REBUTTAL:
I am happy with the authors' response and will therefore increase my score from 6 to 7.

**Time Spent Reviewing:**

2

---

> ### Author Response · Authors · 2021-08-10
> **Response to reviewer QY5N**
>
> Thank you very much for your time and feedback! We are delighted to see that you do not have much to critique and think that our paper should get published. Given this positive outlook, may we suggest that you increase your score to make your commitment more clear? We respond to your different points below.
>
> **Performance on CIFAR:** Please note that crucially, we do not use data augmentation on CIFAR-10 nor batch-normalization. This is mainly motivated by the fact that data augmentation complicates the Bayesian modelling perspective and is thus often omitted in BNN works (e.g. [1]). That being said, we would like to point out that the SOTA performance of a variational Bayesian method on CIFAR-10 without data augmentation is actually around 87% [2], so we get reasonably close to that. We also took care of convergence during training to guarantee that the models are trained to their full potential and lead to consistent results under our experimental conditions. We are also considering adding CIFAR-100 as additional verification.
>
> **Training details:** Thank you for your question. You are right in observing that, in order for our theoretical guarantees to hold, the particles need to be updated simultaneously. That is, each particle needs to be updated once before the next step of updates can be computed. Note that the particles need not necessarily be in memory at the same time for that. One could imagine a setting where the particles are stored on the hard disk and are sequentially loaded in and out of the main memory to compute their respective updates. However, this would probably lead to some memory access overhead in terms of computational time.
>
> That being said, it is an interesting direction for future work to study whether one could train the particles completely sequentially, that is, train each particle to convergence while repelling it from the already training particles. This would probably lead to a technique similar to universal boosting [3]. Following this direction it would also be interesting to see if the methods could benefit from repulsion with past samples, intuitively this is going to increase the number of samples used in the different gradient estimators and might reduce their variance.
>
>
>
> [1] Wenzel, F., Roth, K., Veeling, B. S., Świątkowski, J., Tran, L., Mandt, S., ... & Nowozin, S. (2020). How good is the bayes posterior in deep neural networks really?. arXiv preprint arXiv:2002.02405.
>
> [2] Ober, S. W., & Aitchison, L. (2021, July). Global inducing point variational posteriors for bayesian neural networks and deep gaussian processes. In International Conference on Machine Learning (pp. 8248-8259). PMLR.
>
> [3] Campbell, T., & Li, X. (2019). Universal boosting variational inference. Advances in Neural Information Processing Systems, 32, 3484-3495.

---

> > ### Comment · Reviewer_QY5N · 2021-08-31
> > **Response to Response to reviewer QY5N**
> >
> > I am happy with the authors' response and will increase my score from 6 to 7. Thank you!

---

### Official Review · Reviewer_evs4 · 2021-07-19

**Rating:** 7
**Confidence:** 3

**Summary:**

The paper tackles the problem of building ensemble methods to improve prediction accuracy and uncertainty estimates. The authors propose adding a repulsive term to the gradient update rule for each ensemble element to produce diverse ensembles, and they design repulsive terms for which the empirical distribution over ensemble members converges to the true posterior distribution. Specifically, the authors propose three repulsive terms that arise from three alternative approximations of the gradient of the empirical distribution over ensemble members, and they compare the resulting ensembles with standard deep ensembles in synthetic tasks and image classification tasks. Finally, to account for the fact that different weights can parameterize the same function (and therefore a repulsive term in the weight space may not achieve the desired effect), the authors consider updates (and repulsive terms) in both weight space and function space and compare both approaches in each evaluation task.

The claims are that the proposed approach:
* enables using deep ensembles to perform proper Bayesian inference,
* improves the quality of the uncertainty estimates in synthetic tasks,
* improves OOD detection on image datasets.

**Limitations And Societal Impact:**

The paper proposes a method whose end goal is to improve accuracy and uncertainty prediction across a wide range of machine learning prediction problems. From this perspective, the general benefits and potential negative societal impact of these problems, such as reinforcing certain biases, could be applied here.

**Main Review:**

*Significance*: Deep ensemble methods enable improving prediction accuracy as well as producing better-calibrated uncertainty estimates. The paper proposes introducing a kernel-based repulsive term that improves ensemble performance as well as improves the theoretical motivation for the ensemble, therefore I think the paper is relevant and will be impactful for the community.

*Experiments*:
* The synthetic experiments illustrate how the proposed approaches operating in function space can produce better uncertainty estimates than standard ensembles and weight-space methods. Furthermore, the 2D classification experiment also illustrates how the proposed approaches in function space can produce better estimates than f-SVGD in some OOD regions. The synthetic experiments do not generally show significant differences between the three repulsion terms, though SGE-WGD appears to fit the target better in Figure 2.
* The image experiments show that the proposed approaches generally outperform the standard ensembles, although the ranking is not consistent across evaluation tasks/metrics.
* Overall, the empirical evaluation is not very thorough but it supports the claim that the repulsive terms are beneficial. I think that the section could be improved by 1) more analysis or discussion on the potential trade-offs in using each repulsive term so as to provide guidelines for future use, 2) highlighting how the proposed approach does not plateau as discussed in the abstract.

*Clarity*:
* The paper is generally well written. That said, the plots could use a bit more information, e.g. the colors for the densities at the top and right of each plot in Figure 2 are not described, and the captions do not highlight the take-aways.
* Minor: \beta appears only in line 196 and is not described, seems to be equal to the inverse of the normalizing term in eq (16).

**Time Spent Reviewing:**

4-6

---

> ### Author Response · Authors · 2021-08-10
> **Response to reviewer evs4**
>
>  Thank you very much for your time and feedback! We are happy to see that you found our paper relevant and impactful. We respond to your different points below.
>
> **Different estimators:** We will add a more detailed discussion of the different gradient estimators and their respective tradeoffs in terms of performance and computational cost.
>
> **Plateau with an increasing number of particles:** We agree that it is important to show that our approach does not lead to a plateau when the number of particles increases. In the case of the ResNets on CIFAR, this experiment is slightly beyond the capabilities of our computational infrastructure, nevertheless, we will study this phenomenon for an ensemble of MLPs on fashion MNIST so that the computational cost is still accessible and we will possibly add the results to our analysis.
>
> **Clarity:** Thanks for pointing out our oversights with regards to labelling the colours in Fig. 2 and the $\beta$ term. We will clarify those.

---

### Official Review · Reviewer_jr1C · 2021-07-20

**Rating:** 6
**Confidence:** 3

**Summary:**

The paper proposes a particle based inference method that, unlike Deep Ensembles, approximates the Bayesian posterior of a deep neural network. The proposed approaches are variants of Stein Variational Gradient Descent. They employ a repulsive force (in the form of a kernel) that prevents the particles from collapsing into the same mode.

The main contribution of the work is showing that this particle-based training method is equivalent to gradient descent minimizing the KL-divergence of the true posterior and the particle approximation in Wasserstein space.

The paper also presents empirical results. On a 1d synthetic benchmark, it is shown that the proposed methods are able to capture in-between uncertainty. On a 2d synthetic benchmark, it is shown that the uncertainty estimates are akin to that of HMC. On larger, image classification datasets, it is shown that the methods outperform both deep ensembles and svgd.

**Limitations And Societal Impact:**

They are addressed adequately.

**Main Review:**

### Quality:

The work is good quality. The theoretical results are convincing and the results are interesting and relevant to researchers working on ensembles. It interprets repulsive ensembles as an approximate particle based inference method by showcasing that the proposed update rules correspond to the gradient flow of the KL-divergence in Wasserstein space. This is very interesting as it allows practitioners to use these particle-based methods with the confidence that they approximate the Bayesian posterior.

The theoretical results are well presented in an organized way. Related works are clearly explained and contrasted.

A shortcoming is that the experiments section has almost no detail on the implementation. Basic information, such as the number of particles used, is missing. I was able to locate some of these in the supplementary material, but I believe they should be in the main text. It would not be possible to replicate these results based on the level of detail given here.

The baseline methods slightly underperform. A ResNet 32 should achieve higher than 85% accuracy on Cifar 10.

* From the experiments in Figure 3, it seems that the weight-space methods are all identical to Deep Ensembles. Is this the case because of the high dimensional weight space? That the distances of the particles are too large to be able to affect each other?

### Novelty:

The paper heavily builds on the results of [1], but formulates the update rules slightly differently. To my knowledge, the theoretical results are also novel, although it is possible that I am unfamiliar with related works.

### Presentation:

The motivation and the narrative are presented very well, but I found parts of the paper difficult to understand. Here are some parts that are unclear to me:

* What form does $k(f_1, f_2)$ take in practice? The RBF kernel is mentioned as an example, but is this the one used in the experiments? Is it computed over a batch of training points? How are the kernel hyperparameters determined?
* How is $\log \pi(f)$ computed? Given a weight-space prior, the prior density is difficult to estimate due to the non-identifiability problem.
* What are the advantages and drawbacks of SGE and SSGE? Does one give a more accurate approximation? Are there computational tradeoffs?

[1] Wang, Ziyu, et al. "Function space particle optimization for Bayesian neural networks.”


**Time Spent Reviewing:**

3.5

---

> ### Author Response · Authors · 2021-08-10
> **Response to reviewer jr1C**
>
> Thank you very much for your time and feedback! We are happy to see that you found our results convincing, interesting, relevant, and well-presented. We respond to your different points below.
>
> **Implementation details:** You are right in observing that we deferred most of our implementation details to the appendix. We will remedy this by moving essential details (number of particles, etc.) back to the main text.
> We will also clarify that in the experiments, we always use the RBF kernel and for the function space case compute it over the current minibatch of training data points. We use the popular median heuristic [3] to choose the kernel bandwidth so that no parameters need to be manually tuned. We agree that it is an interesting future direction to explore different kernels and methods for bandwidth selection. Interestingly, following the arguments in [5], where they show that the Manhattan distance metric (L1 norm) is consistently more preferable than the Euclidean distance metric (L2 norm) for high dimensional settings, we performed some new experiments using the L1 norm (Laplace kernel) for the KDE and we will add them together with a discussion. We also noticed in our tests that an adaptive bandwidth is generally overperforming a fixed one. We will mention this in our analysis.
> The gradient of the prior in function space was estimated in all methods using the SSGE estimator. Given that it is the only one that allows to obtain gradient estimates at positions out of the sample points and therefore the samples from the prior can be collected only once avoiding additional computational cost. We will add these details in the main text.
> We will also add a more detailed discussion of the different gradient estimators and their tradeoffs in terms of performance and computational cost. In particular, the latter can be found in the original paper that introduced SSGE [4]; both method SSGE and SGE have a computational cost of $O(M^3 +M^2d)$ with $M$ the number of points and $d$ their dimensionality. SSGE has the additional costs for predictions $O(M(d + J))$ with $J$ the number of eigenvalues.
>
> **Performance on CIFAR:** Please note that crucially, we do not use data augmentation on CIFAR-10 nor batch-normalization. This is mainly motivated by the fact that data augmentation complicates the Bayesian modelling perspective and is thus often omitted in BNN works (e.g. [1]). That being said, we would like to point out that the SOTA performance of a variational Bayesian method on CIFAR-10 without data augmentation is actually around 87% [2], so we get reasonably close to that.
>
> **Performance of weight-space methods:** It is correct and interesting to observe that the weight-space kernel methods do not (at least visually) improve over the standard deep ensembles in the one-dimensional regression task in Fig. 3. We would argue that this can be  due to different factors:
> 1. The non-identifiability of NNs: because of their weight-space symmetries, that is, different weights can implement the same function. This means that the particles can repel each other in the weight space and (in principle) still all implement the same function and thus not actually capture any functional uncertainty. This is our main motivation for introducing functional kernels instead.
> 2.The number of particles: as explained in section 3.2 the convergence to the exact posterior is ensured in the asymptotic limit of infinite particles. Therefore the number of members in the ensemble used in the weight space methods can have an important influence on the predictive posterior.
> 3.The curse of dimensionality: it is known in the literature that kernel methods and so KDE suffer from the curse of dimensionality [6]. Following this, the gradient estimators we are creating are affected by the same issue. This leads to a weight repulsive component that is still beneficial (see results in MNIST and CIFAR) but that, at least in the toy experiments, seems to be unable to recover the true predictive posterior. This effect motivated us to move the inference in the function space where the lower dimensionality can help overcoming this limitation leading to a predictive posterior that better resemble the one obtained with HMC.
>
>
> [1] Wenzel, F., Roth, K., Veeling, B. S., Świątkowski, J., Tran, L., Mandt, S., ... & Nowozin, S. (2020). How good is the bayes posterior in deep neural networks really?. arXiv preprint arXiv:2002.02405.
>
> [2] Ober, S. W., & Aitchison, L. (2021, July). Global inducing point variational posteriors for bayesian neural networks and deep gaussian processes. In International Conference on Machine Learning (pp. 8248-8259). PMLR.
>
> [3] Liu, Q., & Wang, D. (2016). Stein variational gradient descent: A general purpose bayesian inference algorithm. arXiv preprint arXiv:1608.04471.
>
> [4] Jiaxin Shi, Shengyang Sun, and Jun Zhu. A spectral approach to gradient estimation for implicit distributions. In International Conference on Machine Learning, pages 4644–4653. PMLR, 2018.
>
> [5] Aggarwal, Charu C., Alexander Hinneburg, and Daniel A. Keim. "On the surprising behavior of distance metrics in high dimensional space." International conference on database theory. Springer, Berlin, Heidelberg, 2001.
>
> [6] Gramacki, A. (2018). Nonparametric kernel density estimation and its computational aspects. Cham: Springer International Publishing.

---

> > ### Comment · Reviewer_jr1C · 2021-08-17
> > **Thank you for the reply**
> >
> > Thank you for the detailed reply. I am happy with the response and I am keeping my score.

---

### Decision · Program_Chairs · 2021-09-27

**Decision:**

Accept (Spotlight)

**Comment:**

In this paper the authors proposed the addition of a repulsive term to the training objective of deep ensembles, based on Stein variational inference.  They show that this turns the standard deep ensemble training into an approximation to the Bayesian posterior.  The reviewers found the proposed method compelling and found the paper to be of high writing and technical quality.  They also found the work timely and that it addresses an important open problem.  Thus the recommendation is to accept.